# CRISPR-mediated gene correction links the ATP7A M1311V mutations with amyotrophic lateral sclerosis pathogenesis in one individual

Yeomin Yun[1,2,17], Sung-Ah Hong [3,4,17], Ka-Kyung Kim[5], Daye Baek[1,2], Dongsu Lee[6], Ashwini M. Londhe[7,8], Minhyung Lee[9,10], Jihyeon Yu[3], Zachary T. McEachin [11], Gary J. Bassell[11,12], Robert Bowser[13], Chadwick M. Hales[14], Sung-Rae Cho[2,15], Janghwan Kim[9,10], Ae Nim Pae[7,8], Eunji Cheong[6], Sangwoo Kim [5], Nicholas M. Boulis[16], Sangsu Bae [3,4]* & Yoon Ha[1,2]*

Amyotrophic lateral sclerosis (ALS) is a severe disease causing motor neuron death, but a complete cure has not been developed and related genes have not been defined in more than 80% of cases. Here we compared whole genome sequencing results from a male ALS patient and his healthy parents to identify relevant variants, and chose one variant in the X-linked *ATP7A* gene, M1311V, as a strong disease-linked candidate after profound examination. Although this variant is not rare in the Ashkenazi Jewish population according to results in the genome aggregation database (gnomAD), CRISPR-mediated gene correction of this mutation in patient-derived and re-differentiated motor neurons drastically rescued neuronal activities and functions. These results suggest that the ATP7A M1311V mutation has a potential responsibility for ALS in this patient and might be a potential therapeutic target, revealed here by a personalized medicine strategy.

[1] Department of Neurosurgery, Spine & Spinal Cord Institute, College of Medicine, Yonsei University, Seoul 03722, South Korea. [2] Brain Korea 21 PLUS Project for Medical Science, College of Medicine, Yonsei University, Seoul 03722, South Korea. [3] Department of Chemistry, Hanyang University, Seoul 04763, South Korea. [4] Research Institute for Natural Sciences, Hanyang University, Seoul 04763, South Korea. [5] Department of Biomedical Systems Informatics, Yonsei University College of Medicine, Seoul 03722, South Korea. [6] Department of Biotechnology, College of Life Science and Biotechnology, Yonsei University, Seoul 03722, South Korea. [7] Convergence Research Center for Diagnosis, Treatment and Care System of Dementia, Korea Institute of Science and Technology, PO Box 131, Cheongryang, Seoul 130-650, South Korea. [8] Division of Bio-Medical Science & Technology, KIST School, Korea University of Science and Technology, Seoul 02792, South Korea. [9] Stem Cell Convergence Research Center, Korea Research Institute of Bioscience and Biotechnology (KRIBB), Daejeon 34141, South Korea. [10] Department of Functional Genomics, KRIBB School of Bioscience, Korea University of Science and Technology, Daejeon 34113, South Korea. [11] Laboratory of Translational Cell Biology, Emory University School of Medicine, Atlanta, GA 30322, USA. [12] Department of Cell Biology, Emory University, Atlanta, GA 30322, USA. [13] Department of Neurobiology, Barrow Neurological Institute and St. Joseph's Hospital and Medical Center, Phoenix, AZ 85013, USA. [14] Department of Neurology, Emory University, Atlanta, GA 30322, USA. [15] Department and Research Institute of Rehabilitation Medicine, Yonsei University College of Medicine, Seoul 03722, South Korea. [16] Department of Neurosurgery, Emory University School of Medicine, Atlanta, GA 30322, USA. [17]These authors contributed equally: Yeomin Yun, Sung-Ah Hong. *email: sangsubae@hanyang.ac.kr; HAYOON@yuhs.ac

DNA sequencing technologies have evolved in high throughput manners to generate tremendous volume of data inexpensively, since the development of Sanger sequencing in 1977[1]. Now, personalized whole genome information can easily be obtained via next-generation sequencing (NGS)[2,3]. Such advances suggest that individualized treatment might be possible in genetic disorders, such as neurodegenerative diseases[4]. In previous studies, rare disorder-linked mutations were discovered using family based whole genome or exome sequencing for autism, Alzheimer's disease, ALS, and Parkinson's disease[5–10]. In biological research, pathogenicity in individual patient can be determined via in vitro disease modeling using induced pluripotent stem (iPS) cells derived from the patient, which reflect full genetic information of the patient[11–15]. Furthermore, programmable nucleases, including CRISPR-Cas (clustered, regularly interspaced short palindromic repeats, and CRISPR-associated) nucleases, enable single nucleotide substitutions very accurately at desired target sites[16–19]. Through targeted gene editing of known disease variant in patient-derived iPS cells via CRISPR-Cas nucleases, pathologic metabolisms, such as signaling pathway, transcripts change, and functional rescue of a gene have been investigated for Huntington's disease, cardiac disease, and ALS[20–22].

Amyotrophic lateral sclerosis (ALS; or Lou Gehrig's disease), an adult onset neurodegenerative disease, is characterized by the loss of motor neurons (MNs), muscle weakness, and death from respiratory failure within 3 years from the onset of symptoms for half of all patients[23,24]. Most cases are sporadic; only about 10% of individuals with ALS have a familial history. Since mutations in the superoxide dismutase 1 (SOD1) gene were first revealed to cause ALS disease[25], mutations in over 20 other genes have been linked to ALS. Until now, variants on SOD1 gene have deeply been studied from patient-derived iPS cells using CRISPR-Cas9 nucleases. Comparison studies between MNs differentiated from ALS patient-derived iPS cells and MNs from SOD1 gene corrected cell lines revealed that functional changes by SOD1 mutations induce neurodegeneration and aberrant gene expression, resulting in vulnerable oxidative stress, altered protein response pathway, ER stress, and mitochondrial defect[26–28]. In more than 80% of ALS cases, however, the responsible genes for ALS still remain unclear[29–31]. Therefore, there is an important need to elucidate additional ALS-relevant genes from ALS patients.

Because of rapid improvements in DNA sequencing, iPS cells, and gene editing technologies, investigations of the relationship between genotype and phenotype in each patient can be realized. In this study, we combined the whole genome sequencing (WGS) and CRISPR-Cas9 mediated gene editing technologies to identify disease linked mutations in an ALS patient, as well as to verify the disease relevance of the mutation in patient-derived iPS cells, suggesting the strategy of personalized medicine.

## Results

**Trio-based WGS identifies responsible variants in a patient**. In the case of a patient of Jewish origin who was diagnosed with bi-brachial type sporadic ALS disease in his sixties, we tried to identify susceptibility genes for personalized ALS treatment. Because the patient had no mutations in the previously reported genes related to ALS, we carried out a trio-based WGS study of this patient and his healthy parents to identify genetic mutations causing ALS. The whole workflow of WGS data analysis is summarized in Fig. 1, and detailed pipelines are described in Supplementary Fig. 1. The trio WGS data were analyzed based on pedigree information in X-linked, recessive, or de novo genetic mode; we excluded the possibility of autosomal dominant inheritance because the patient's parents were healthy. We

annotated called variants in each inheritance mode, and prioritized according to the order of severity from Sequence Ontology (SO) and the degree of pathogenicity from the American College of Medical Genetics and Genomics (ACMG) guidelines[32]. Common variants frequently observed in panels of the normal population were filtered out. Next, we evaluated functional importance and prioritized following two workflows in parallel. First, we selected variants with rare allele frequencies (<0.1%) in the Ashkenazi Jewish population that were predicted to be functionally high impact based on the order of severity as defined by SO (Supplementary Table 1). Among 21 selected variants, we prioritized 7 that were located in functional gene networks using both direct (Supplementary Table 2) and indirect (Supplementary Table 3) network-prioritization methods based on 126 ALS-related genes registered in the ALS genetic database[29] (Supplementary Table 4). Second, we selected possible pathogenic variants using two in silico analysis tools, Polyphen-2 and SIFT, according to the ACMG guidelines. We evaluated the pathogenic impact of each variant on the function and structure of the corresponding protein and then prioritized 2 variants with higher pathogenic scores (Supplementary Table 5). From the above two independent criteria, we obtained 9 candidate variants that were possibly pathogenic in ALS.

We excluded four of the nine variants because of predictions that they were related to conditions other than neurodegenerative disease and because they were located in non-functional domains of the genes (Supplementary Table 6). For the remaining five variants, we performed genotyping to determine a zygosity of each gene variant using Sanger sequencing (Supplementary Fig. 2). The results revealed that variants in DTX1 and NOTCH2 were false positive; i.e. each site only contained wild type sequence rather than frameshift or missense variant that was expected from WGS. And the variants in GFRA2 and GART were identified to be heterozygous; i.e. one allele in each site had wild type sequence. On the other hand, ATP7A located on the X chromosome had a mutated variant only likewise to WGS data, identified to be hemizygous because the patient is male[33]. Throughout the prioritization process, we selected the ATP7A variant as a strong candidate for causing the disease. The ATP7A protein, a transmembrane P-type ATPase, has a well-characterized role in maintaining intracellular copper (Cu) concentrations, which are essential for regulating the activity of Cu-dependent enzymes that function in oxidation/reduction reactions, signaling pathways, and metabolism[34–37].

In conclusion, the filtering steps described above indicate that the M1311V mutation in the ATP7A gene may be causative in the etiology of ALS (Fig. 1). The ATP7A gene is X-linked and we found that the mother of the ALS patient in question was a carrier of the M1311V mutation (Supplementary Fig. 3). Our selection is further supported by previous studies showing that abnormal Cu homeostasis increases the production of reactive oxygen species (ROS), resulting in cellular damage, neurodegeneration, and neuroinflammation[35–37], and by other studies showing that several ATP7A variants are linked with neurodegenerative diseases such as Menkes disease, occipital horn syndrome, and X-linked distal spinal muscular atrophy type 3, which show similar symptoms as ALS[37] (Supplementary Fig. 4).

**CRISPR-mediated gene correction in patient-derived iPS cells**. The best way to prove correlation between the ATP7A M1311V mutation and the ALS disease in the patient is to compare the phenotype of patient-derived neurons containing the ATP7A M1311V mutation against that of corrected neurons, because the two cell types would have identical backgrounds except for the ATP7A mutation (Fig. 2a). To generate ATP7A-corrected cell

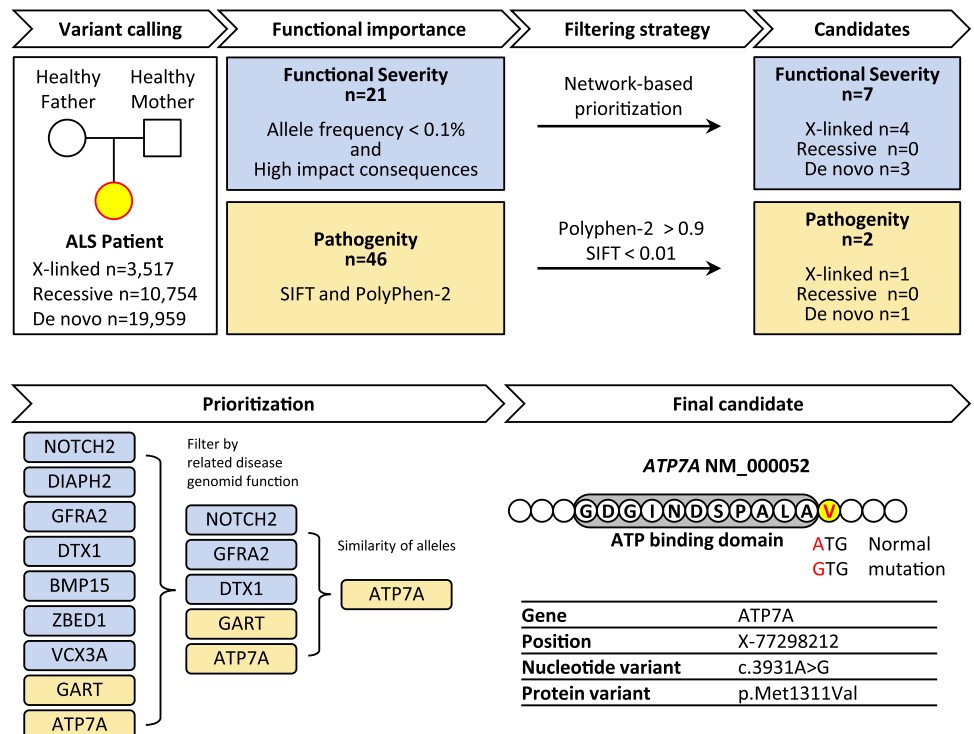

**Fig. 1 Filtering strategy to identify pathogenic mutation.** Trio whole genome sequencing was performed for the patient and his healthy parents to annotate inherited and de novo mutations. Candidate variants were prioritized according to measures of functional importance including allele frequency, high impact consequences, or in silico analysis. Variants were further prioritized using network-based prioritization or pathogenity scores. Nine variants were determined to be high priority; this list was narrowed down by review of related diseases, predicted genomic function, and selection of candidates with zygosity analysis. The final candidate mutation, ATP7A M1311V, was selected after a review of literature for neurodegenerative disease.

lines, we first constructed iPS cells, named ATP7A-M1311V, from the patient's fibroblasts. Then, we transfected CRISPR-Cas9 nuclease into the iPS cells along with single-stranded oligodeoxynucleotides (ssODNs) as the donor DNA for gene correction (Fig. 2b and Supplementary Fig. 5). We initially measured a correction rate of 0.4% from the bulk iPS cells via targeted deep sequencing[38], so we repetitively divided them into small populations of cells. After five repeated clump passages, we ultimately obtained two completely corrected iPS cell lines (Fig. 2c), named ATP7A-Cor1 and ATP7A-Cor2.

To investigate the side effects by CRISPR-Cas9, we examined the mutations at predicted potential off-target sites that had 1–3-nt mismatches or 1-nt DNA or RNA bulges using Cas-OFFinder software[39] in the two corrected cell lines. We performed targeted deep sequencing for 14 sites but detected no significant off-target effects (Supplementary Tables 7–9). In addition, we checked whether the gross organization of the chromosomal DNA of the ATP7A-Cor lines was damaged during the gene editing process using karyotyping. Both ATP7A-Cor lines showed normal karyotype morphologies, as did ATP7A-M1311V (Fig. 2d). Sanger sequencing was performed to confirm the gene corrected sequence in differentiated cell type, neural progenitor cells (NPCs) and MNs (Fig. 2e), indicating that CRISPR-mediated gene correction gave no substantial effects to the iPS cells.

**Gene correction of the ATP7A M1311V leads to improvement.** To compare the pathologic phenotype and neuronal function of the ATP7A-M1311V and ATP7A-Cor lines, we next differentiated the ATP7A-M1311V, ATP7A-Cor1, and ATP7A-Cor2 lines to neural progenitor cells (NPCs) and measured lysyl oxidase (LOX) activities and ROS levels in each case. LOX is a Cu-dependent enzyme that is activated by Cu transported by

ATP7A[40]. Compared to the ATP7A-M1311V-NPC line, ATP7A-Cor1-NPC and ATP7A-Cor2-NPC showed increased LOX activities and produced lower ROS levels that were comparable to those from control NPCs from a healthy donor (Supplementary Fig. 6). These results support the premise that *ATP7A* gene correction may improve ATP7A function and ALS pathology in NPCs.

Given this evidence, we further differentiated one of the corrected NPC lines, ATP7A-Cor1-NPC, as well as ATP7A-M1311V-NPC and the control NPCs, to MNs (Fig. 3a), as follows. NPCs were initially purified using a cell surface marker, polysialic acid-neural cell adhesion molecule (PSA-NCAM), to eliminate non-neuronal cells from cell population of neural rosettes[41]. Most PSA-NCAM positive NPCs consistently showed to retain high level of neural cell markers, Nestin and Tuj1 in each cell line (Fig. 3b, c). NPCs were differentiated in motor neuron differentiation medium for 3 weeks and ultimately exhibited a motor neuron positive marker, Isl1/2 (Fig. 3d, e). To determine whether gene correction of ATP7A-M1311V affect neurite complexity, we measured the expression level of a neuronal marker, microtubule associated protein 2 (MAP2), in each line by immunostaining and also visualized nuclei by 4′, 6-diamidino-2-phenylindole (DAPI) staining. The quantitative differentiation ratios calculated by comparison of the DAPI and MAP2 positive areas indicated that ATP7A-M1311V-MN showed reduced MAP2 level compared with ATP7A-Cor1-MN, which showed similar efficacy as the control line (Fig. 3f, g), implying declined regulating of neuritic maintenance and microtubule networks in dendritic morphologenesis[42–44] in the ATP7A-M1311V line and recovery from this effect in the ATP7A-Cor1 line.

For each set of differentiated MNs, we measured the concentration of released lactate dehydrogenase (LDH), a measure of cell damage. LDH release was higher in ATP7A-

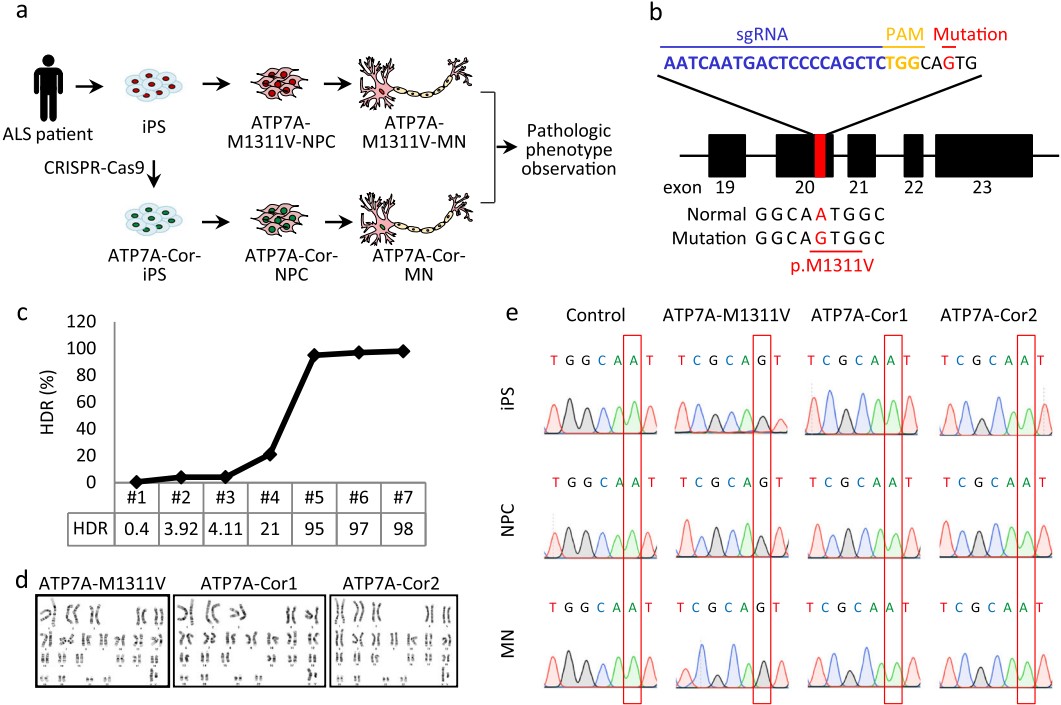

**Fig. 2 CRISPR-mediated gene correction of the ATP7A M1311V mutation and construction of patient-derived isogenic iPS cell lines. a** Overview of our personalized medicine approach. **b** Strategy for correction of the ATP7A M1311V mutation in patient-derived iPS cells using CRISPR-Cas9. The ATP7A point mutation (A to G) is shown in red. The 20-nucleotide single guide RNA (sgRNA) target sequence and the protospacer adjacent motif (PAM) are shown in blue and yellow, respectively. The single-stranded oligodeoxynucleotides (ssODNs), which has 50-bp homology arms, was designed to change the disease-causing mutation to the normal sequence and disrupt the PAM sequence to prevent further Cas9-mediated cleavage after the correction. **c** Efficiency of homology-directed repair (HDR) for generating the gene-corrected isogenic iPS cell line. **d** Karyotype analysis of parental (ATP7A-M1311V) and gene-corrected (ATP7A-Cor1 and ATP7A-Cor2) iPS cell lines. **e** Sanger sequencing to confirm the mutant ATP7A M1311V and normal sequence in iPS cells, NPCs, and MNs.

M1311V-MN than in the control MNs, whereas release was significantly decreased in ATP7A-Cor1-MN, comparable to the level in the control MNs (Fig. 3h). We next measured the cell viability and ROS levels after treatment with glutamate, which acts as an excitatory neurotransmitter to generate action potentials. Although glutamate plays an essential role in the nervous system, it can also exert a neurotoxic effect, causing free radical production, mitochondrial dysfunction, and protein aggregation, especially in ALS-derived MNs. Cell viability measurements showed that only 80% of ATP7A-M1311V-MN cells survived under glutamate treatment compared to the untreated condition, whereas most ATP7A-Cor1-MN cells survived, similar to the control MNs, indicating rescue of the glutamate vulnerability in ATP7A-Cor1-MN (Supplementary Fig. 7). The ROS products significantly accumulated in ATP7A-M1311V-MN cells under basal conditions (Fig. 3i) and when they were treated with glutamate (Supplementary Fig. 7), whereas the ROS level was reduced in ATP7A-Cor1-MN cells, similar to that in the control MNs, also supporting the concept of cell function recovery in ATP7A-Cor1-MN. To further confirm whether mitochondria-related function was rescued in the corrected MN lines, we measured mitochondrial membrane potentials linked to the ROS generation using tetramethylrhodamine ethyl ester (TMRE) assay. In line with above results of ROS, ATP7A-M1311V-MN cells showed impaired mitochondrial membrane potentials, whereas the TMRE level was recovered in ATP7A-Cor1-MN cells similar to the control MNs (Fig. 3j). Taken together, we confirmed the differences in cell viability, cellular function and neurite complexity between ALS patient-derived cells and ATP7A gene-corrected cells.

**Improvement of neuronal activity in gene-corrected MNs.** To further compare neuronal properties, 8 weeks after MN differentiation (Fig. 4a), we performed electrophysiological analyses for each single neuron cell and four types of action potential (repetitive, adaptive, single, and no firing) were defined and classified (Fig. 4b). Compared to the control MN population, in which most cells showed firing action potentials, half of the ATP7A-M1311V-MN cells showed non-firing action potentials. However, to our surprise, most ATP7A-Cor1-MN cells showed firing action potentials and the relative proportions of the firing patterns were very similar to those of the control MNs (Fig. 4c). We found that ATP7A-M1311V-MN cells showed the loss of voltage gated sodium current and had smaller voltage gated potassium currents compared to the ATP7A-Cor1-MN cells and the control MNs (Fig. 4d, e and Supplementary Fig. 8), which explains why half of the ATP7A-M1311V-MN cells showed non-firing action potentials. These results strongly suggest the recovery of neuronal activity in ATP7A-Cor1-MN compared to ATP7A-M1311V-MN cells.

One potential concern about our results is that the pluripotency and differentiation of iPS cells might have affected the function of the resulting MNs. To address this issue, we constructed induced neural stem cells (iNSCs) from the patient's fibroblasts (ATP7A-M1311V-iNSC) via direct conversion (Supplementary Fig. 9) and compared them with control iNSCs from a healthy donor[45]. We found that the ATP7A-M1311V-iNSC population showed no difference in the proliferation rate compared to the control iNSCs, but showed lower cell viability, lower LOX activity, and elevated ROS under basal cell conditions, similar to the results from the experiments with iPS cells

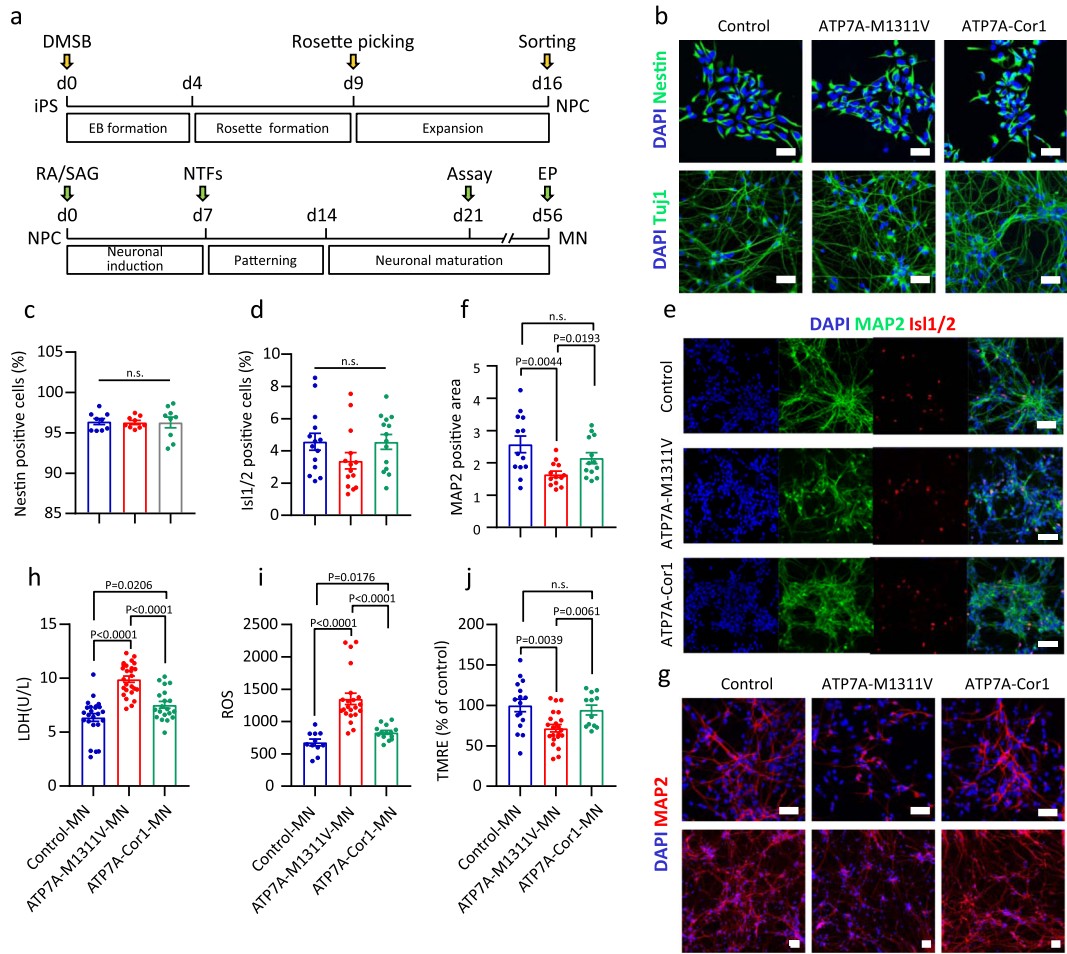

**Fig. 3 Gene correction of the ATP7A M1311V mutation results in improvement of pathophysiology. a** Time table showing differentiation protocol used for NPC from patient-derive iPS and MN from NPC. DMSB, Dorsomorphin and SB431542; EB, Embryoid body; RA, Retinoic acid; SAG, Smoothened Agonist; NTFs, Neurotrophic factors; EP, Electrophysiology. **b** Immunostaining with the anti-Nestin antibody in NPCs and anti-Tuj1 antibody after 3 weeks differentiation. Scale bars, 50 µm. **c** The Nestin-positive cells normalized to that of the 4′, 6-diamidino-2-phenylindole (DAPI)-positive area. $n = 9$. **d** The Isl1/2-positive cells normalized to that of the 4′, 6-diamidino-2-phenylindole (DAPI)-positive area. $n = 14$. **e** Immunostaining with the anti-isl1/2 antibody and anti-MAP2 antibody after 3 weeks differentiation. Scale bars, 50 µm. **f** The microtubule associated protein 2 (MAP2)-positive area normalized to that of the 4′, 6-diamidino-2-phenylindole (DAPI)-positive area. $n = 13$. **g** Immunostaining with the anti-MAP2 antibody (red). DAPI (blue) was used for nuclear staining. Scale bars, 50 µm. **h** LDH activity in motor neurons (MNs). $n = 24$ (Control-MN), $n = 26$ (ATP7A-M1311V-MN), and $n = 20$ (ATP7A-Cor1-MN). **i**, Quantification of the reactive oxygen species (ROS) level. $n = 11$ (Control-MN), $n = 22$ (ATP7A-M1311V-MN), and $n = 14$ (ATP7A-Cor1-MN). **j**, Quantification of the mitochondrial membrane potential using tetramethylrhodamine ethyl ester (TMRE) assay. $n = 16$ (Control-MN), $n = 22$ (ATP7A-M1311V-MN), and $n = 12$ (ATP7A-Cor1-MN). Data are presented as mean ± s.e.m,; unpaired two-tailed t test with Welch's correction. Source data used for the graphs in (**c**, **d**, **f**) and (**h**–**j**) can be found in Supplementary Data 1.

described above (Supplementary Fig. 10), supporting our conclusion that the ATP7A M1311V mutation has a potential responsibility for ALS.

## Discussion
In this study, we revealed that the X-linked ATP7A M1311V mutation has a potential responsibility for ALS in one male ALS patient by comparing WGS results with his healthy parents. Considering various factors including genetic mode, mutation type, allele frequency, and published literature, it is acceptable that ATP7A M1311V has high functional impact on neurodegenerative disease. ATP7A plays a role in Cu transport and maintenance of Cu homeostasis that is critical for proper neurological function such as axonal outgrowth, synapse integrity, and neuronal activation[37]. Homology modeling of ATP7A, which consists of transmembrane regions (MA, MB, and M1-M6) and cytosolic N-, P-, and A-domain, indicates that M1311 residue is located next to the ATP binding site in core cytosolic P-domain,

which induces conformational changes during copper transport process. The M1311 residue has hydrophobic interaction with residues on P- and A-domain, and also has hydrogen interaction with residues on P-domain. In homology model with M1311V mutation, the mutated residue seems to break hydrophobic contacts or hydrogen bond interaction depending on the orientation of valine and surrounding residues (Supplementary Fig. 11). From the molecular modeling study, we hypothesized that ATP7A M1311V mutation disturbs internal interactions between residues on A- or P-domain, and thereby negatively affects the function of Cu binding and its transporting of ATP7A proteins[46–49], which may cause neurodegenerative mechanism in the patient. Alternatively, our results showed that the gene correction of ATP7A M1311V dramatically improves the neuronal morphology and function, which strongly supports the hypothesis.

To date, many researchers have worked to reveal interactions between genetic variants and disease phenotypes. Recently,

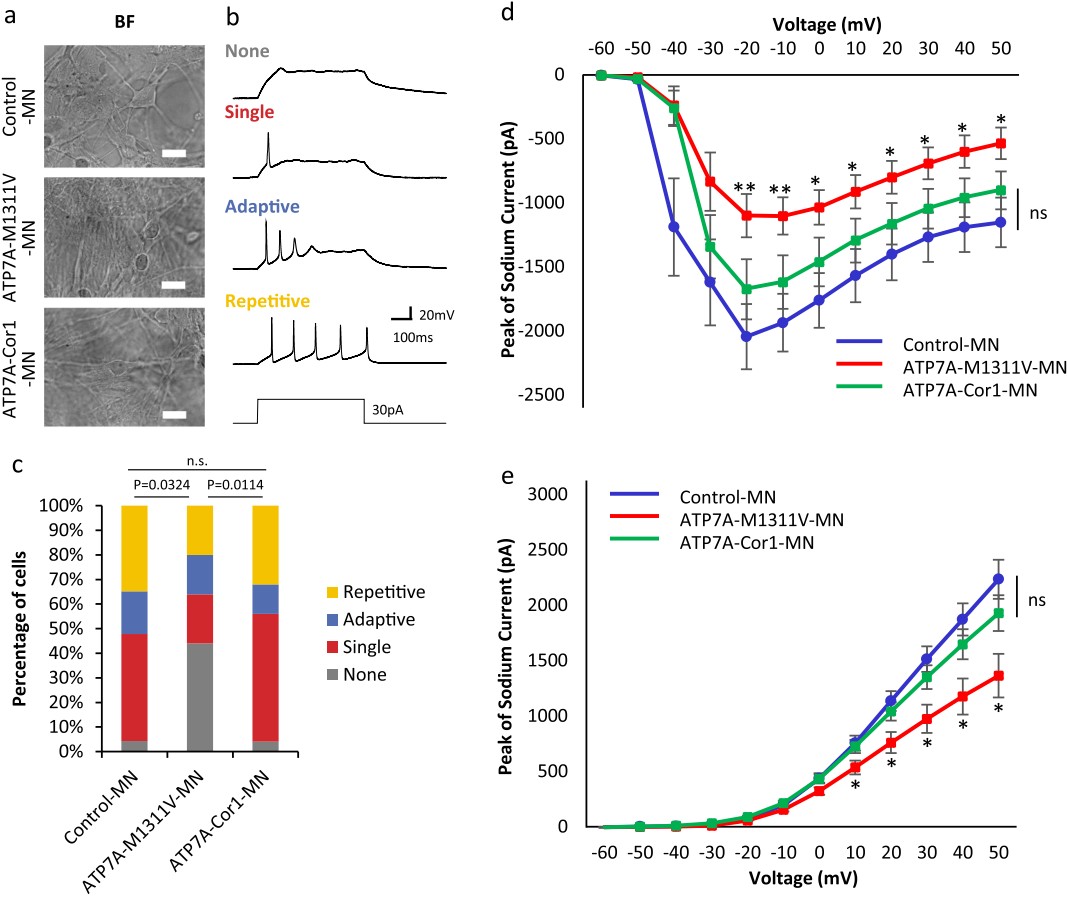

**Fig. 4 Action potential and current traces of voltage-gated sodium and potassium channels in gene-corrected MNs. a** Morphology of neural precursor cells (NPCs) after differentiation for 8 weeks for electrophysiology experiments. BF, bright field. Scale bars, 20 μm. **b** Representative traces of different types of action potentials: none, single, adaptive, repetitive. Repetitive firing was defined as action potentials that lasted for the duration of the current injection (1 s), whereas adaptive firing was defined as multiple action potentials that ceased before the end of the current injection. Comparisons between ATP7A-M1311V MNs and the other groups of cells were performed using data from recordings of firing cells. **c** Proportion of cells firing in each category as a measure of functional maturation. $n = 23$ (Control-MN), $n = 25$ (ATP7A-M1311V-MN), and $n = 25$ (ATP7A-Cor1-MN). The differences between groups were assessed using Fisher's exact test with Bonferroni correction. **d** Sodium currents of Control-MN ($n = 23$), ATP7A-M1311V-MN ($n = 25$), and ATP7A-Cor1-MN ($n = 30$). Every data point was compared with the peak of the sodium current for ATP7A-M1311V-MN. **e** Potassium currents of Control-MN ($n = 23$), ATP7A-M1311V-MN ($n = 25$), and ATP7A-Cor1-MN ($n = 30$). Every data point was compared with the peak of the potassium current for ATP7A-M1311V. The error bars indicate ± s.e.m.; statistical analysis was performed using two tailed $t$-test. $*p < 0.05$, $**p < 0.01$. Source data used for the graphs in (**c–e**) can be found in Supplementary Data 2.

genome-wide quantification analyses have shown that rare mutations are found in normal tissue, whereas risk variants from neurodevelopmental disorder patients are common genetic variants, implying that non-rare genetic variants may cause genetic disease in some patients[50–55]. In this study, we showed that correction of the ATP7A M1311V mutation can rescue the function of MNs derived from cells of one ALS patient of Ashkenazi Jewish descent. However, the ATP7A M1311V allele frequency is not rare in Ashkenazi Jews (0.01991), whereas it is rare in other ethnic populations (0–0.0003) such as European, Asian, Latino, and African. Because ALS is known to be a multifactorial disease and over 100 genes are intricately involved[29,56–58], the *ATP7A* gene may act as one crucial component in an ALS-related network and the ALS disease may be caused by a combination of other disease factors such as genetic defects of other variants in a patient' genome, altered transcripts, and epigenetic and environmental factors together with ATP7A M1311V[59–62]. In our results of trio-based WGS, the patient had many genetic variants in addition to the variants in *ATP7A*. Although we prioritized the ATP7A M1311V mutation as a strong candidate for ALS pathogenesis because of its higher pathogenic score and the X-

linked recessive mutation, we could not exclude other variants completely, which might be a further study. From one deep individual study, we suggest that a non-rare genetic variant in an ALS patient has a potential to be responsible for the disease. Furthermore, in the case of this ALS patient, ATP7A M1311V might be a potential therapeutic target, elucidated by our personalized medicine strategy.

## Methods

**Human samples**. This study includes one trio family (affected patient and unaffected parents) of Ashkenazi Jewish descent. Tissue sample was obtained by Gary Bassell and Chadwick Hales and the healthy donor sample was kindly provided by Zachary T. McEachin at Emory University. The Institutional Review Board (IRB) of Severance Hospital approved this study (IRB approval number: 4-2012-0028).

**Whole genome sequencing (WGS)**. Steps in the WGS analysis for the unaffected parents and the affected patient, such as alignment, variant calling, annotation, and candidate variant selection, are summarized below and detailed in Supplementary Fig. 1. Genomic DNA was extracted from blood. A library of DNA fragments was constructed using a TruSeq PCR-free Library Preparation Kit (Illumina). Paired-end sequencing of the library was performed using rapid run v2.0 (Illumina) chemistry on a GiSeq 2500 (Illumina) sequencer according to the manufacturer's recommended protocol. The genome was sequenced to a mean read depth of <30X.

Raw FASTQ format sequences were aligned to the hg19 reference genome using BWA-MEM and generated to BAM file format. PCR duplicates were removed, known alleles were marked (dbsnp version 138, b37), and defined intervals were targeted for local realignment, after which indels were realigned with default options. We then additionally filtered for aligned files with MQRankSum (Z-score from Wilcoxon rank sum test of Alt vs. Ref read mapping qualities) < −12.5, ReadPosRankSum (Z-score from Wilcoxon rank sum test of Alt vs. Ref read position bias) < −8.0, QD (Quality of Depth) < 2.0, FS (Phred- scaled $p$-value using Fisher's exact test to detect strand bias) > 60.0 or FS > 200, and MQ (RMS Mapping Quality) < 40.0. After filtering, the following numbers of single nucleotide variants (SNVs) remained: 4,667,802 for the affected patient, 4,641,608 for the unaffected father, and 4,643,959 for the unaffected mother.

Mutations were called from the alignment files using GATK haplotype caller following GATK Best Practices recommendations. The VCF files produced by the pipeline use reference allele based on the positive strand of hg19 in the REF field, and variants were shown in the ALT field.

Relatedness was tested for the trio by comparing single nucleotide polymorphism (SNP) concordance between the patient and parents using vcftool. We carried out variant analysis based on pedigree information. X-linked variants were selected if they were present in the patient and mother, but not in the father. De novo variants were chosen if the variants were only present in the patient and not in either parent or by using the TrioDeNovo program. After this analysis, 3517 X-linked variants, 10,754 autosomal recessive variants, and 19,959 de novo variants remained. We annotated each variant type using snpEff and panels of normal populations such as the 1000 Genome Project (1KGP) and the Exome Aggregation Consortium (ExAC).

**WGS data analysis**. We selected functionally important variants with high or moderate impacts in SO. For variants in coding regions, we selected those with functional importance, such as missense, nonsense, and splicing variants. We next selected rare variants by excluding alleles with a global minor allele frequency > 5% in dbSNPv135, a global minor allele frequency in 1KGP > 2%, a minor allele frequency in ExAC > 5%, and a minor allele frequency in the Ashkenazi Jewish population > 5% in gnomAD r2.0.2.

We then followed the guidelines of the American College of Medical Genetics to prioritize variants. Those variants passing the pathogenicity predictions were annotated using SIFT and PolyPhen-2 scores. We used two algorithms to predict the impact of protein sequence variants for missense variants. SIFT is based on sequence homology and the physical properties of amino acids, and PolyPhen-2 is based on eight sequence and three structure predictive features. Possibly damaging SNVs (PDSs) were extracted from these datasets. We then reviewed existing literature on neurodegenerative diseases, protein structures, and pathways. These steps yielded information on a variant's functional importance, predicted the resulting damage to protein function, and indicated biologically relevant pathways. In addition to prioritization, we manually checked for alignment quality with Integrated Genomic Viewer (IGV, https://www.broadinstitute.org/software/igv) and cBioPortal (http://www.cbioportal.org/mutation_mapper.jsp).

**Network-based gene prioritization**. To prioritize the selected genes with functionally important variants, we performed a network-based gene prioritization method that propagates functional scores via a network model. There are two approaches for network propagation, which are direct neighbor and network diffusion. The former can transmit the functional score only by direct links in the network model. The representative algorithm is naïve Bayes[63], in which the score of a particular node label is the sum of the network edge weights of all of the connected neighbors for the same label. The latter can diffuse the functional score throughout the entire network, typically by Gaussian smoothing (GS)[64]. The GS algorithm finds an optimal solution in which it achieves a minimal difference between the initial and final scores of a labeled gene, and also between the label score of a gene and each of its neighbors.

In our study, we first chose the STRING v10.5[65] database as the base network model. STRING v10.5 includes direct (physical) interactions, as well as indirect (functional) interactions such as co-expression, shared selective signals across genomes, text-mining of the scientific literature, and transfer of interaction knowledge between organisms based on gene orthology. Using high-confidence (confidence score >0.7) links of STRING v10.5 and genes known to be linked with amyotrophic lateral sclerosis (ALS) (Supplementary Table 3) collected from the database[29], we prioritized the ALS candidate genes by both network propagation approaches, and selected ALS candidate genes with functionally important variants (Supplementary Tables 4 and 5).

**Induced neural stem cell (iNSC) direct reprogramming**. Reprogramming into iNSCs was performed using a previously described method[45] with slight modifications. Briefly, Sendai reprogramming vectors (CytoTune-iPS 2.0 Sendai Reprogramming Kit, Thermo Fisher Scientific) were transduced into 30,000 fibroblast cells according to the manufacturer's instructions. The next day, the culture medium was replaced with a neural reprogramming medium (RepM-Neural: Advanced DMEM/F12 and Neurobasal medium were mixed at a ratio of 1:1 and supplemented with 0.05% AlbuMAX-I, 1× N2, 1× B27 minus vitamin A,

2 mM GlutaMAX, 0.11 mM β-mercaptoethanol, all from Thermo Fisher Scientific), and cells were cultured for 18–21 days. During reprogramming, a chemical cocktail including 0.2 mM NaB (Sigma), 3 μM CHIR99021 (Tocris), and 0.5 μM A83-01 (Tocris) was added to the medium before use. The resulting colonies were manually picked and maintained in a RepM-Neural medium containing 3 μM CHIR99021, 0.5 μM A83-01, and 10 ng ml$^{-1}$ hLIF (Sigma).

RNA extraction and quantitative PCR were performed as previously described[66] using a Power SYBR Green Kit (Applied Biosystems). The primer sequences were as follows: 5′-TCTTTGCTTGGGAAATCCG-3′ (PAX6 forward), 5′-CTGCCCGTTCAA CATCCTTAG-3′ (PAX6 reverse), 5′-TGGCTAAGAACATCGGAAGG-3′ (SeV forward), 5′-GTTTTGCAACCAAGCACTCA-3′ (SeV reverse).

**ATP7A homology model generation**. Discovery studio 2018 software (Dassault Systèmes BIOVIA, Discovery Studio Modeling Environment, Release 2018, San Diego: Dassault Systèmes, 2018.) was used for our protein structure modeling study. An ATP7A homology model was created using the 2.75 Å X-ray crystal structure of P1B-ATPase (E2P state) from Legionella pneumophila (PDB ID 4BBJ)[46]. The template structure (4BBJ) has transmembrane (TM) helices and cytosolic N-, P-, and A-domains but excludes heavy metal binding domains. This structure was obtained in copper-free E2P state after Cu was transported through the TM domain. The ATPase_human sequence (Q04656) was aligned to the template, and it showed an 18.7% sequence identity and 30.0% sequence similarity. Initially, 20 homology models were created and validated using the DOPE score and Ramachandran plot. Validated structures were then selected for methionine residue interaction analysis. Ten models for the ATP7A M1311V mutant were then created, and the effects of the mutation on interactions between residues were analyzed. Images were rendered using PyMOL software (www.pymol.org) and Discovery Studio 2018 software.

**Construction and evaluation of single guide RNAs (sgRNAs)**. All sgRNAs were designed using Cas-Designer (http://www.rgenome.net/cas-designer/), and two complementary oligos encoding each sgRNA were annealed and ligated with a plasmid used for sgRNA expression, which had been linearized with BsaI (New England Biolabs). Oligos were purchased from Macrogen Inc (South Korea). A list of oligos used to generate sgRNAs is provided in Supplementary Fig. 5.

Patient-derived fibroblasts were grown in Dulbecco's Modified Eagle Medium (DMEM, WELGENE) with 10% fetal bovine serum (FBS) (WELGENE), 0.2% penicillin/streptomycin mix (Gibco), 1x MEM NEAA (Gibco), and 55 μM β-mercaptoethanol (Sigma). 150,000 patient-derived fibroblasts were co-transfected with 500 ng of Cas9-encoding plasmid and 500 ng of sgRNA-encoding plasmid using a Neon™ Transfection System (Invitrogen) with the following parameters: voltage, 1600; width, 20 ms; number, 1. Three days after transfection, the cells were harvested, and genomic DNA was isolated using NucleoSpin Tissue (MACHEREY-NAGEL & Co. KG) according to the manufacturer's protocol. The genomic DNA was amplified and analyzed by targeted deep sequencing.

**Establishment of gene-corrected isogenic iPS cell lines**. The iPS cell lines from the patient and a healthy donor were established by Zachary T. McEachin using CytoTune-iPS Sendai Reprogramming Kits. To produce the normal *ATP7A* sequence, patient-derived iPS cells were transfected with plasmids encoding Cas9 and sgRNA targeting the ATP7A M1311V mutation with the appropriate single-stranded oligodeoxynucleotides (ssODNs) as a donor DNA. On day 3, transfected iPS cells were detached with ReLeSR™ (Stem Cell Technologies) and split for clump passage accompanied by serial dilution. When colonies containing round-shaped iPS cells were formed at around 7–14 days, they were detached using ReLeSR™. A small proportion of cells from colonies was harvested for targeted deep sequencing, and the leftover cells were transferred for further expansion. To prepare genomic DNA, cell pellets were resuspended in 50–100 μl of Proteinase K extraction buffer [40 mM Tris-HCl (pH 8.0) (Sigma), 1% Tween-20 (Sigma), 0.2 mM EDTA (Sigma), 10 mg of proteinase K, 0.2% nonidet P-40 (VWR Life Science)], incubated at 60 °C for 15 min, and heated to 98 °C for 5 min, and the genomic DNA was amplified and analyzed by targeted deep sequencing. Colonies with high frequencies of homology-directed repair (HDR) were picked for repeated clump passage accompanied by serial dilution and genotyping. As the HDR frequency was increased through this process, colonies containing the normal *ATP7A* sequence were established. Karyotyping was conducted at Gendix, Inc (Seoul, South Korea).

**Genotyping**. Hundred nanogram of genomic DNA or 2–3 μL of Proteinase K extraction solution was amplified by three-round PCR using SUN-PCR blend (SUN GENETICS). The following programs were used for the first round PCR: 3 min at 95 °C; 25 cycles of 30 s at 95 °C, 30 s at 60 °C, and 60 s at 72 °C; and, finally, 3 min at 72 °C. 1 μL of the first round PCR product was used for the second round PCR. The following programs were used: 3 min at 95 °C; 25 cycles of 30 s at 95 °C, 30 s at 60 °C, and 20 s at 72 °C; and, finally, 3 min at 72 °C. 1 μL of the second round PCR product was used for the third round PCR. The following programs were used: 3 min at 95 °C; 30 cycles of 30 s at 95 °C, 30 s at 62 °C, and 20 s at 72 °C; and, finally, 3 min at 72 °C. The amplified genomic DNA regions containing a Cas9-targeted site were purified using Expin™ PCR SV mini (GeneAll)

according to the manufacturer's protocol. The PCR amplicons were sequenced using a MiniSeq Sequencing System (Illumina), and the results were analyzed using Cas-Analyzer (http://www.rgenome.net/cas-analyzer/). A list of primers for PCR is provided in Supplementary Fig. 6.

**Cell culture and neural differentiation of iPS cells.** The iPS cells were cultured in mTeSR™1 (Stem Cell Technologies) medium on hESC-qualified Matrigel (Corning)-coated culture dishes. To induce embryoid body (EB)-mediated neural rosette formation, iPS cells were harvested using 2 mg ml⁻¹ collagenase IV (Gibco) and EBs were suspension cultured for 4 days with 5 μM SB431542 (SB) (Tocris) and 5 μM Dorsomorphin (DM) (Tocris) in TeSR™-E6 (Stem Cell Technologies) medium. On day 5, EBs were attached to Matrigel-coated culture dishes containing DMEM/F12 (Gibco) supplemented with 1× N2 (Gibco), 20 ng ml⁻¹ basic fibroblast growth factor (bFGF) (R&D system), and insulin (Sigma) and grown for an additional 5 days. Neural rosettes that appeared in the center of EBs were isolated using a pulled glass pipette and cultured on Matrigel-coated culture dishes in neural progenitor cell (NPC) medium (DMEM/F12 medium supplemented with 1× N2, 1× B27 (Gibco), 20 ng ml⁻¹ bFGF (R&D system) and 20 ng ml⁻¹ epidermal growth factor (EGF) (Peprotech)). This medium was replaced every 2 days for the following week. PSA-NCAM positive NPCs were used to motor neuron differentiation. NPCs were seeded at low density on Matrigel-coated culture dishes in differentiation medium (NPC medium deprived of bFGF and EGF) with 1 μM retinoic acid (RA) (Sigma) and 0.5 μM Smoothened Agonist (SAG) (Millipore). On day 7 of differentiation, RA and SAG were withdrawn and the medium was changed to differentiation medium plus 0.1% fetal bovine serum (FBS) (Hyclone), 200 μM ascorbic acid (Sigma), 10 ng ml⁻¹ each of brain derived neurotrophic factor (BDNF), glial-derived neurotrophic factor (GDNF), and ciliary neurotrophic factor (CNTF) (Peprotech). For further maturation, differentiated cells were maintained in differentiation medium with 0.1% FBS. All cell lines were differentiated in parallel and compared at the same time.

**Isolation of PSA-NCAM positive cells.** Neural rosette cells were expanded to a confluence of 80–90% and treated with 10 μM Y27632 (Stem Cell Technologies) for 1 h to prevent cell death. After dissociation with Accutase (Gibco), the cells were pelleted, resuspended in a solution containing 0.5% bovine serum albumin (BSA) (Gibco) in 1x phosphate buffered saline (PBS), and incubated on ice for 10 min. Anti-PSA-NCAM Microbeads (Miltenyi Biotec) were added to the solution, which was then incubated on ice for 15 min. Cells were washed in 1× PBS and loaded on a LS column for magnetic separation. The column was washed with 1× PBS, causing unlabeled cells to pass through, after which the plunger was pushed into the column to flush out magnetically labeled cells.[41]

**Sanger sequencing validation of the *ATP7A* sequence.** The 5 mutation variants (*DTX1, NOTCH2, GFRA, GART*, and *ATP7A*) were confirmed by Sanger sequencing. The genomic DNA was extracted from the ALS-patient derived fibroblasts and GM14867 using NucleoSpin Tissue (MACHEREY-NAGEL & Co. KG) according to the manufacturer's protocol, and 100 ng of the genomic DNA were amplified using KOD -Multi & Epi- (TOYOBO) under the following conditions: 2 min at 94 °C; 35 cycles of 10 s at 98 °C, 30 s at 60 °C, and 20 s at 68 °C; and, finally, 2 min at 68 °C. The primer sequences were as follows: 5′-CCTGTGCTGAACCTCATGC-3′ (*DTX1* forward), 5′-TGAGCGGGTAGGCGA GGT-3′ (*DTX1* reverse), 5′-TGCTGAGGTCCTTGGAGAGT-3′ (*NOTCH2* forward), 5′-ACACCCACTACCTCCTGTG-3′ (*NOTCH2* reverse), 5′-GTTCAGCA GAGGGAATCTGG-3′ (*GFRA2* forward), 5′-ACCCTATAGCCTGCAAAGCA-3′ (*GFRA2* reverse), 5′-TCTCCCTTCTCAGGATCGAA-3′ (*GART* forward), 5′-TTCTGGGATGGGTGTTCACT-3′ (*GART* reverse), 5′-TGATAGATGCTGCCA GATGC-3′ (*ATP7A* forward), and 5′-GCTGGGTTGAGAAAGTCAGG-3′ (*ATP7A* reverse). The amplified genomic DNA was purified using Expin™ PCR SV mini (GeneAll) or Expin™ Gel SV mini (GeneAll) according to the manufacturer's protocol. Sanger sequencing were performed at Macrogen.

The *ATP7A* sequence was validated by Sanger sequencing of genomic DNA that was extracted from iPS cells, NPCs, and MNs using a Genomic DNA Extraction Kit (Bioneer) and amplified by PCR using the following conditions: 5 min at 95 °C; 35 cycles of 30 s at 95 °C, 30 s at 58 °C, and 60 s at 72 °C; and, finally, 10 min at 72 °C. The primer sequences were as follows: 5′-ATTGCTCAGTTATGTTTCACGTACTC-3′ (*ATP7A* forward) and 5′-TCTTAATGGCTGATAGCATGGAACT-3′ (*ATP7A* reverse). Sanger sequencing experiments were performed at Macrogen Inc.

**Electrophysiology.** Whole-cell patch-clamp recordings were used to investigate the functionality of ALS neurons. The ALS neurons were differentiated for 8 weeks and transferred to a recording chamber, and perfused with aerated (95% O₂/5% CO₂ mixed gas) artificial cerebrospinal fluid bath solution containing 124-mM NaCl, 3-mM KCl, 1.3-mM MgSO₄, 1.25-mM NaH₂PO₄, 26-mM NaHCO₃, 2.4-mM CaCl₂·2H₂O, and 10-mM glucose at 32 °C. Patch pipettes (open pipette resistance, 2–5 MΩ) were filled with an internal solution consisting of 140-mM K + -gluconate, 10-mM NaCl, 1-mM CaCl₂, 10-mM HEPES, 0.2-mM EGTA, 5-mM Mg-ATP and 0.5-mM Na-GTP. For confirmation test, 0.5-μM tetrodotoxin and 5-mM tetraethylammonium were used (Supplementary Fig. 9). ALS neurons were visualized with Olympus upright BX51WI microscope with 60× immersion lens.

Patch electrodes were pulled on Sutter P-97 horizontal puller (Sutter Instrument Company) from borosilicate glass capillaries (World Precision Instruments, Sarasota, FL). Recorded signals were amplified using MultiClamp 700B amplifier (Axon Instruments, Union City, CA), and data acquisition was performed using Digitizer 1550B and pClamp10 software (Axon Instruments). Intrinsic membrane properties were investigated using voltage clamp mode, and firing properties were investigated using current-clamp mode. To measure sodium and potassium currents, cells were stimulated with 100-ms voltage steps of depolarizing current starting from −60 mV to +20 mV (+10 mV per step). In current-clamp mode, cells were subjected to a series of current injections to examine action potential generation. Only cells with a series resistance that was less than 20 MΩ, and a resting membrane potential (RMP) that was more hyperpolarized than −20 mV were included in data analysis.

**LDH assay.** To measure released LDH, MNs were differentiated from NPCs as described above for 3 weeks. The cell culture medium was transferred to 1.5-ml microcentrifuge tubes, and centrifuged at 2000 rpm for 3 min to remove cell debris, after which the supernatant was transferred to new 1.5-ml microcentrifuge tubes. Released LDH was measured using a commercially available LDH assay kit (Abcam). Sample absorbance was measured at OD 450 nm, and LDH concentration was calculated according to the manufacturer's instructions.

**Treatment of cells with copper and glutamate.** iNSCs were treated with CuSO₄·5H₂O (Sigma) to examine the effects of Cu on ROS levels and LOX activity. Cells were seeded, incubated for 24 h, and then exposed to Cu at various concentrations. After 4 h, cells were washed with Dulbecco's PBS (DPBS), and ROS and LOX assays were performed. MNs were treated with 1-mM glutamate (Sigma) to test glutamate vulnerability and effects of glutamate on ROS levels. After 1 h, cells were washed and MTT and ROS assays were performed, or cells were immunostained with antibodies recognizing cleaved caspase-3.

**MTT assay.** The 3-[4,5-dimethylthiazol-2-yl]-2,5 diphenyl tetrazolium bromide (MTT) assay was used to measure glutamate-induced cell toxicity. Cells were seeded in 96-well plates, incubated for 24 h, and then treated with 1-mM glutamate for 1 h. Cells were then washed with DPBS (Gibco), and MTT assay was performed. The 0.5 mg ml⁻¹ MTT solution (Sigma) was added to each well, and plates were incubated at 37 °C for 4 h. The reaction was stopped by removing MTT solution and adding dimethyl sulfoxide to dissolve the formazan crystals. VersaMax ELISA Microplate Reader (Molecular Device) was used to read the absorbance at 560 nm.

**Cell viability assay.** To quantify cell viability, live and dead cells were stained using a LIVE/DEAD Cell Viability/Cytotoxicity Kit (Thermo Fisher Scientific) in accordance with the manufacturer's protocol. Briefly, calcein AM was used to stain live cells and ethidium homodimer-1 was used to stain dead cells. Cells were exposed a mixture of the two dyes and incubated at room temperature for 15 min. Cells were counted using live images randomly captured by fluorescence microscopy, and analyzed using Image J software.

**Immunocytochemistry.** MNs and iNSCs were fixed with 4% paraformaldehyde at room temperature for 10 min and washed with 0.3% Tween 20 in PBS. Cells were blocked with 10% normal donkey serum, 0.3% Triton-X in PBS at room temperature for 1 h, and incubated with primary antibodies at 4 °C overnight. Cells were then washed with 0.3% Tween 20 in PBS, and incubated with secondary antibodies (Jackson Immunoresearch) at room temperature for 30 min. Images were acquired using an LSM-700 confocal microscope (Carl Zeiss). Primary antibodies recognizing the indicated proteins were obtained as follows: MAP2 (Abcam), cleaved caspase-3 (Cell Signaling Technology), Tuj1 (Abcam), Ki67 (ZYMED, Abcam), PAX6 (DSHB), Sox2 (R&D Systems), Nestin (Santa Cruz), PLZF and N-CAD (Millipore); 4′, 6-diamidino-2-phenylindole (DAPI) (Santa Cruz Biotechnology) and Hoechst (Millipore) were used for nuclear staining. For further analysis of neuronal morphology, Image J was used to study cells stained with anti-MAP2 antibodies and DAPI.

**Reactive oxygen species (ROS) measurement.** ROS levels were measured in live cells using 2′,7′-dichlorofluorescin diacetate (DCFDA) (Sigma). The iNSCs and NPCs were seeded a day before the assay was performed, while MNs were differentiated from NPCs for 3 weeks beforehand. Cells were exposed to 1-μM DCFDA for 30 min at 37 °C. Fluorescence indicative of ROS was measured with at Ex/Em = 485/535 nm, and the fluorescence level was normalized O.D. unit.

**Lysyl oxidase (LOX) assay.** The iNSCs and NPCs were seeded a day before the assay was performed, while MNs were differentiated from NPCs 3 weeks beforehand. Cell samples were harvested and lysed with 200-μl lysis buffer. LOX activity was determined using a commercially available LOX assay kit (Abcam) according to the manufacturer's protocol. Fluorescence was detected at Ex/Em = 540/590 nm

and normalized to the protein concentration measured using BCA kit (R&D Systems).

**Mitochondrial membrane potential assay.** MNs were differentiated from NPCs for 3 weeks beforehand. A day prior to assay, MNs were re-seeded in 96-well plates at a density of $2 \times 10^4$. Mitochondrial membrane potential was determined by cytofluorimetric analysis after staining with tetramethylrhodamine ethyl ester (TMRE) (Abcam) at concentration of 1 μM for 30 min at 37 °C. Fluorescence indicative of TMRE was measured with at Ex/Em = 510/580 nm.

**Statistical analysis and Reproducibility.** Values are expressed as means ± s.e.m., and the differences between groups were assessed using unpaired two-tailed t test with Welch's correction, and Fisher's exact test and proportion test with Bonferroni correction. Multiple t test with correction for multiple comparisons using the Holm-Sidak method and Bonferroni-Dunn method was used for multiple comparisons. P values less than 0.05 were considered significant. Statistical analysis was performed with GraphPad Prism7 (San Diego, CA, USA). In cases of using NPCs and MNs, all experiments were biologically replicated across 2 independent neural differentiations from iPSCs. That is, from one corrected iPSC line (ATP7A-Cor1), we repeatedly differentiated it to neural progenitor cells (ATP7A-Cor1-NPCs) and further motor neurons (ATP7A-Cor1-MNs). In case of using iNSCs, all experiments were reproduced at least 4 times. All attempts for replication were successful.

**Reporting summary.** Further information on research design is available in the Nature Research Reporting Summary linked to this article.

## Data availability

Targeted deep sequencing data have been deposited in the NCBI Sequence Read Archive database (SRA; https://www.ncbi.nlm.nih.gov/sra) under accession number PRJNA531568.

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

## Acknowledgements

The authors appreciate the family for their participation in this study. The authors thank J.M. Lee and C. Justin Lee at Center for Cognition and Sociality, Institute for Basic Science for providing valuable comments. This work was supported by the Gluck Family Foundation. This work was also supported by National Research Foundation of Korea (NRF) Grants (no. 2018M3A9H3022412 to S.B., no. 2015M3A9C7030128 to J.K., no. 2013R1A1A2062110 to K.K.K., no. 2015R1D1A1A02059821 to Y.H.) and by a grant from the Korea Healthcare technology R&D Project (HI16C1012) to S.B and by a faculty research grant from the Yonsei University College of Medicine (6-2018-0161) to Y.H.

## Author contributions

N.M.B. and Y.H. conceived the project. Y.Y., Y.H. and S.B. designed the experiments. Y.Y., S.A.H., J.Y. and D.B. performed experiments and interpreted data. D.L. and E.C. performed all electrophysiological studies. K.K.K. and S.K. analyzed genetic information. Z.T.M., R.B., M.L. and J.K. provided cell lines. G.J.B. and C.M.H obtained the IRB protocol number for Emory fibroblast collection and providing the healthy donor cell line. A.M.L. and A.N.P performed homology model generation. S.R.C. obtained the IRB protocol number for Yonsei University. Y.Y. and S.B. wrote the manuscript. All of the authors discussed the results and commented on the manuscript.

## Competing interests

The authors declare no competing interests.
