## [Peer Review File · Communications Biology]

Reviewers' comments:

Reviewer #1 (Remarks to the Author):

The authors claim that X-linked ATP7A M1311V variant is linked to ALS pathology. They use whole genome sequencing for analysis of ALS linked mutations in the patient compared to his healthy parents.

The data generation and enrichment in DNA seq analysis seems adequate however, following concerns should be addressed by the authors prior to the publication.

Major Concerns:

1. The authors claim that ATP7A M1311V mutation is responsible for ALS which contradicts what they also say: "this variant is not rare in the Ashkenazi Jewish population according to results in the genome aggregation database (gnomAD)"

2. After computational analysis, they find 5 candidate genes and then filter them into two, GART and ATP7. They then, excluded GART because of a paper they cited showed GART is not causing FALS. However, in the same paper, they clearly say GART must be considered as a potential candidate. if authors would correct the mutant variant of GART in the IPS cells, would they get similar results?

3. Similar techniques on IPS cell has to be employed to test the GART variant.

4. If conducting new experiments is not possible at least the paper should be rewritten focusing on findings, all potential candidate genes in this patient, and showing ATP7A M131V experiment in IPS cells as a case study plus a validation of their method. Thus, they can also conclude that other candidate genes might have contributed to the disease (Especially GART) in different levels according to their WGS analysis.

5. The nature of the sporadic ALS cases are not fully understood, it is quite possible that these several candidate genes might have relationship with the ALS in the patient combined.

The genomic sequencing and computational analysis of the study are strong and highlights on these possible candidates, which overlap with what is known in the literature. Therefore, this study gives us highly valuable outcomes using family-based genome sequencing. However, the authors' conclusion to say ATP7A-M1311V is solely responsible for ALS in this patient is highly disputable. Even the authors acknowledge that the ALS might be combination of other factors and ATP7A-M1311V combined in the last paragraph of the discussion. However, they should emphasize on their findings by WGS suggesting the potential roles of the other candidate genes. GMP6B, PAGE1, NOTCH2 and GART can be emphasized.

Minor Issues:

1. References 5-7 are not particular to family-based sequencing studies for the particular diseases mentioned. They are review papers. In this current format, it misleads the reader.

2. This sentence below is troublesome.

"The best way to prove causation between the ATP7A M1311V mutation and the ALS disease in the patient is to compare the phenotype of patient-derived neurons containing the ATP7A M1311V mutation against that of corrected neurons, because the two cell types would have identical backgrounds except for the ATP7A mutation"

Correction of mutation in IPS cells do not explain causation but establish a link, the authors' should

have introduced the patient mutation into a WT iPSC cell line and compare the phenotype. Please rephrase whole sentence.

3. Figure 3, in most of the experiments only one corrected cell line is used, could it be clarified?

If they will not repeat the assays with the second cell line please just use the figure 3 c-h

How the n number was determined in each assay?

4. Map2 staining seems to be working but bigger pictures with higher resolution needed to be shown

5. Did the authors check CHAT positive neurons as mature motor neuron marker? Map2 is not a specific motor neuron specific marker.

6. LDH and ROS levels are still significantly different from WT in corrected cells, what can be the reason for this? If the gene correction is not completely rescuing the cells, then would that be possible that the other candidate variants found in the study might be playing a role?

7. Figure 4 motor neuron images are not in good quality. In 8-weeks of cultures what percent viability do the authors observe in mutant motor neurons versus controls?

Reviewer #2 (Remarks to the Author):

This manuscript by Yu et al. reports the discovery of a novel mutation in the gene ATP7A as causal for ALS in a single individual. Following positive clinical identification the authors used deep sequencing approaches for the patient as well as his non-diseased parents to identify M1311V as the point mutation in ATP7A that most likely caused ALS. To confirm this finding, the authors carried out further in vitro analysis of patient-derived motor neurons and demonstrated viability, glutamate sensitivity, oxidative stress and electrophysiological phenotypes. All these phenotypes could be reversed by CRISPR correction mediated reversion of this point mutation. Together, these data provide reasonable likelihood that in this patient a point mutation in ATP7A is causal for ALS. The study is carried out very competently, written clearly and relates current findings with what is known about the genetic basis for ALS. Overall, I don't have any major issues with the study and recommend publication.

Reviewer #3 (Remarks to the Author):

Yun et al. describe a novel mutation associated with ALS by genome sequencing a trio (mother, father and child). Using a combination of in silico prediction of the mutations effect on protein function and literature survey, they prioritize mutations in two genes and follow one of these mutations for functional validation. They first generated a single iPSC line from the patient's fibroblasts and then used CRISPR-Cas9 mediated genome editing to generate isogenic controls. By employing survival and functional assays, they show that the mutation in ATP7A is detrimental to neuronal function. Based on the deleterious effect of this mutation on neurons, they conclude that the identified mutation is causative of ALS.

The functional assays are well characterized though I have grave concerns on their iPSC derived neurons that they claim as motor neurons. Additionally, I am unsure regarding their use of replicates. Finally, I am not convinced that this mutation is causative of ALS. However, these findings will be of interest to the ALS community, increasing the list of genes that are found to be associated with the disease.

Experimental details regarding the assays are adequate though it would help researchers reproduce their work if details regarding number of neurons used and the time point of the assay is mentioned in

the text or methods section.

1. Fig 3c and 3d. Decreased numbers of MAP2+ neurons is not an indication of degeneration. High variability between iPSC differentiations, even isogenics, is not uncommon. Additionally, MAP2 is not a specific marker for motor neurons. The authors need to show MN efficiency using either ISL1/CHAT or HB9/CHAT staining. MN survival should be followed over a time course of degeneration. Another problem is the relatively large number of non-neuronal cells in the population, likely due to the use of FBS in the differentiation protocol. This can be a major confounding factor for downstream analysis related to survival and apoptosis. If the ALS line has a larger number of non-neuronal cells compared to the controls, it's hard to conclude that the LDH and other whole well assays specifically relate data on MN health. The authors need to mention at what day in the differentiation process the cells were analysed as this can affect the results. Ideally, you want to conduct your assays when your MN are functional.

2. The n= values are a bit puzzling. When the authors mention n=20, are they really saying they did 20 independent differentiations? Or is the data an average of 20 wells? Are these technical replicates or biological? If these are independent differentiations, why do the numbers vary so widely? All 3 lines must be differentiated in parallel and compared at the same time.

3. Since ALS affects MN, why have the authors chosen to analyse iNPCs instead of iNs? Without further characterization of the iNPC population, it is not advisable to use these for functional assays. Do the authors know whether the generated iNPCs are functional i.e. can they differentiate into neurons? Is the efficiency similar between control and ALS fibroblasts? What is the significance of increasing LOX activity in response to Cu? Why was this done only for ALS iNPC?

4. The genetic data does not confirm that the ATP7A mutations causes ALS. Unless this can be confirmed in an independent cohort, we can only say this is likely associated with ALS.

5. Since the ATP7A mutation is not rare in Ashkenazi Jews and the authors claim this mutations as a cause of ALS, wouldn't we expect to find a higher incidence of ALS cases in this population. One possibility is that both alleles need to be mutated to cause ALS suggesting a recessive mode of inheritance. This will be in line with what the authors suggest that the mutation causes a loss of function. The authors need to support this line of inquiry by performing ATP7A1 knockdown and show impairment of MN function and/or survival.

6. Table S4 not explained well.

7. Fig. 2c not explained in the text.

8. Fig S4C. I am guessing mutation in the PAM is synonymous but this should be mentioned.

9. Please clarify what you mean by "5 repeated clump passages".

10. Why was Cor2 dropped from further analysis? Is due to the fact that it showed higher ROS levels?

Reviewer #1 (Remarks to the Author):

The authors claim that X-linked ATP7A M1311V variant is linked to ALS pathology. They use whole genome sequencing for analysis of ALS linked mutations in the patient compared to his healthy parents.

The data generation and enrichment in DNA seq analysis seems adequate however, following concerns should be addressed by the authors prior to the publication.

Major Concerns:

1. The authors claim that ATP7A M1311V mutation is responsible for ALS which contradicts what they also say: “this variant is not rare in the Ashkenazi Jewish population according to results in the genome aggregation database (gnomAD)”

Response: The reviewer raised a critical point of our study. When we corrected the ATP7A M1311V mutation using CRISPR in a specific manner in patient-derived and re-differentiated motor neurons, neuronal activities and functions were drastically rescued, indicating that the ATP7A M1311V mutation is linked to ALS. However, as the reviewer mentioned, this variant is not rare in the Ashkenazi Jewish population according to gnomAD. Because ALS is known to be a multifactorial disease and over 100 genes are intricately involved, we concluded that the ATP7A M1311V mutation can act as one crucial component in an ALS-related network and the ALS disease may be caused by a combination of other disease factors together with ATP7A M1311V at least for this patient. Recent studies [references 51-56] reported that common genetic variants can be risk variants in neurodevelopmental disorder patients, whereas rare mutations are found in normal tissue, supporting our conclusion that non-rare genetic variants can also be a critical factor for genetic disease in a ALS patient. Overall, to address the reviewer’s concern, we have toned down the conclusion from ‘ATP7A M1311V mutation is responsible for ALS’ to ‘ATP7A M1311V mutation has a potential responsibility for ALS’ in the revised manuscript.

2. After computational analysis, they find 5 candidate genes and then filter them into two, GART and ATP7. They then, excluded GART because of a paper they cited showed GART is not causing FALS. However, in the same paper, they clearly say GART must be considered as a potential candidate. if authors would correct the mutant variant of GART in the IPS cells, would they get similar results?

Response: To address the reviewer’s concern, we additionally performed Sanger sequencing for the 5 candidate variants selected from WGS results including GART and ATP7 mutations (Fig. S2). As a result, it revealed that variants in DTX1 and NOTCH2 were false positive; i.e. each site only contained wild type sequence, and the variants in GFRA2 and GART were identified to be heterozygous; i.e. one allele in each site had wild type sequence. On the other hand, ATP7A located on the X chromosome had a mutated variant only, identified to be hemizygous because the patient is male. Therefore, we put the ATP7A variant as a strong candidate for causing ALS with a more confident. However, we cannot exclude the potential role of the GART mutation in ALS completely, thus we have clearly

described it in discussion part (**lines 303-311**) and generally toned down the role of the ATP7A variant. We believe that the investigation of other genes' mutations including the GART mutation would be a following study.

3. Similar techniques on IPS cell has to be employed to test the GART variant.

Response: As described above (#2), we rated the ATP7A mutation at higher priority mainly because of the Sanger sequencing data. Nonetheless, because we cannot exclude the potential role of the GART mutation in ALS completely, we generally toned down the role of the ATP7A variant overall in the revised manuscript. We believe that the investigation of other gene mutations including the GART mutation would be a following study, which takes another long time more than 1 year.

4. If conducting new experiments is not possible at least the paper should be rewritten focusing on findings, all potential candidate genes in this patient, and showing ATP7A M131V experiment in IPS cells as a case study plus a validation of their method. Thus, they can also conclude that other candidate genes might have contributed to the disease (Especially GART) in different levels according to their WGS analysis.

Response: As the reviewer suggested, we tried to tone down the role of the ATP7A M1311V and to describe that our finding is deduced from one case study, which suggests a personalized medicine approach in discussion (**line 310**).

5. The nature of the sporadic ALS cases are not fully understood, it is quite possible that these several candidate genes might have relationship with the ALS in the patient combined. The genomic sequencing and computational analysis of the study are strong and highlights on these possible candidates, which overlap with what is known in the literature. Therefore, this study gives us highly valuable outcomes using family-based genome sequencing. However, the authors' conclusion to say ATP7A-M1311V is solely responsible for ALS in this patient is highly disputable. Even the authors acknowledge that the ALS might be combination of other factors and ATP7A-M1311V combined in the last paragraph of the discussion. However, they should emphasize on their findings by WGS suggesting the potential roles of the other candidate genes. GMP6B, PAGE1, NOTCH2 and GART can be emphasized.

Response: Thank you for the valuable comment. We agree with the reviewer's comment that the ATP7A M1311V mutation is not the only responsible factor for ALS disease. However, additional Sanger sequencing data for the 5 candidate variants (**Fig. S2**) put a higher priority to ATP7A mutation and gene correction of this mutation in patient-derived motor neurons drastically rescued neuronal activities and functions, strongly indicating that the ATP7A M1311V mutation is linked to ALS at least for the patient. Taken together, we have tried to emphasize the combination effects among variants, rather than the sole role of the ATP7A

variant. (especially in **lines 302-306**).

Minor Issues:

1. References 5-7 are not particular to family-based sequencing studies for the particular diseases mentioned. They are review papers. In this current format, it misleads the reader.

Response: To address the reviewer's comment, we have replaced the **References 5-10** to others showing family-based WGS/WES in specific diseases.

2. This sentence below is troublesome.

"The best way to prove causation between the ATP7A M1311V mutation and the ALS disease in the patient is to compare the phenotype of patient-derived neurons containing the ATP7A M1311V mutation against that of corrected neurons, because the two cell types would have identical backgrounds except for the ATP7A mutation"

Correction of mutation in IPS cells do not explain causation but establish a link, the authors` should have introduce the patient mutation into a WT IPS cell line and compare the phenotype. Please rephrase whole sentence.

Response: Thank you for the comment. To address the reviewer's comment, we have changed the word to make it clear from 'causation' to 'correlation' in the sentence (**line 140**).

3. Figure 3, in most of the experiments only one corrected cell lines is used, could it be clarified?

If they will not repeat the assays with the second cell line please just use the figure3 c-h

How the n number was determined in each assay?

Response: As the reviewer mentioned, we initially obtained two completely corrected iPS cell lines, named ATP7A-Cor1 and ATP7A-Cor2, and further differentiated them to neural progenitor cells, named ATP7A-Cor1-NPCs and ATP7A-Cor2-NPCs. Once we confirmed increased LOX activities and lower ROS levels in both NPC lines compared to control, we chose one cell line (ATP7A-Cor1-NPC) and further differentiated it to motor neurons (ATP7A-Cor1-MNs), repeatedly. In **Figure 3**, the n number was determined from independent ATP7A-Cor1-MNs. We have clearly described it in the revised manuscript (**Lines 183-184**). Furthermore, to avoid a misunderstanding, we moved another cell line data (ATP7A-Cor2-NPC) to **Figure S7**.

4. Map2 staining seem to be working but bigger pictures with higher resolution needed to be

shown

Response: To address the reviewer's comment, we have added bigger pictures with higher resolution in lower panel of **Fig. 3g**.

5. Did the authors check CHAT positive neurons as mature motor neuron marker? Map2 is not a specific motor neuron specific marker.

Response: Thank you for the comment. In the revised manuscript, we used a Isl1/2 marker to verify the motor neuron differentiation (**Figs. 3d and 3e**). Actually, we used MAP2 to determine the neuritic maintenance and microtubule networks in dendritic morphogenesis.

6. LDH and ROS levels are still significantly different from WT in corrected cells, what can be the reason for this? If the gene correction is not completely rescuing the cells, then would that be possible that the other candidate variants found in the study might be playing role?

Response: Thank you for the valuable comment. We can suggest two possible reasons as follows. 1) As the reviewer suggested, it might be due to other variants revealed from the trio WGS, because ALS is known to be a multifactorial disease. 2) We used one healthy donor cell that have different genetic background with the patient, resulting in difference between WT and gene corrected line. Thus, we would emphasize that comparing patient-derived cells against gene corrected cells is most valuable in our study.

7. Figure 4 motor neuron images are not in good quality. in 8-weeks of cultures what percent viability do the authors observe in mutant motor neurons versus controls?

Response: **Figure 4a** were taken at relatively high resolution (600x) to show single neuron used for electrophysiology, which results in not-good quality in bright field, but no viability differences between groups was observed in the microscopy. We confirmed the properties and values for capacitance, input resistance, series resistance, and resting membrane potential (RMP) in **Figure S9** and **Table S7**.

Reviewer #2 (Remarks to the Author):

This manuscript by Yu et. al. reports the discovery of a novel mutation in the gene ATP7A as causal for ALS in a single individual. Following positive clinical identification the authors used deep sequencing approaches for the patient as well as his non-diseased parents to identify M1311V as the point mutation in ATP7A that most likely caused ALS. To confirm this finding, the authors carried out further in vitro analysis of patient-derived motor neurons and

demonstrated viability, glutamate sensitivity, oxidative stress and electrophysiological phenotypes. All these phenotypes could be reversed by CRISPR correction mediated reversion of this point mutation. Together, these data provide reasonable likelihood that in this patient a point mutation in ATP7A is causal for ALS. The study is carried out very competently, written clearly and relates current findings with what is known about the genetic basis for ALS. Overall, I don't have any major issues with the study and recommend publication.

Response: We would like to thank this reviewer for the positive comments and support.

Reviewer #3 (Remarks to the Author):

Yun et al. describe a novel mutation associated with ALS by genome sequencing a trio (mother, father and child). Using a combination of in silico prediction of the mutations effect on protein function and literature survey, they prioritize mutations in two genes and follow one of these mutations for functional validation. They first generated a single iPSC line from the patient's fibroblasts and then used CRISPR-Cas9 mediated genome editing to generate isogenic controls. By employing survival and functional assays, they show that the mutation in ATP7A is detrimental to neuronal function. Based on the deleterious effect of this mutation on neurons, they conclude that the identified mutation is causative of ALS.

The functional assays are well characterized though I have grave concerns on their iPSC derived neurons that they claim as motor neurons. Additionally, I am unsure regarding their use of replicates. Finally, I am not convinced that this mutation is causative of ALS. However, these findings will be of interest to the ALS community, increasing the list of genes that are found to be associated with the disease.

Experimental details regarding the assays are adequate though it would help researchers reproduce their work if details regarding number of neurons used and the time point of the assay is mentioned in the text or methods section.

1. Fig 3c and 3d. Decreased numbers of MAP2+ neurons is not an indication of degeneration. High variability between iPSC differentiations, even isogenics, is not uncommon. Additionally, MAP2 is not a specific marker for motor neurons. The authors need to show MN efficiency using either ISL1/CHAT or HB9/CHAT staining.

Response: Thank you for the valuable comment. In this study, we initially obtained two completely corrected iPS cell lines, named ATP7A-Cor1 and ATP7A-Cor2, and further differentiated them to neural progenitor cells, named ATP7A-Cor1-NPCs and ATP7A-Cor2-NPCs. To minimize the variability issue of iPSC, NPCs were sorted by PSA-NCAM (**reference 42** and **method 11**) to eliminate non-neuronal population from rosettes. Then, for one cell line (ATP7A-Cor1-NPCs), we further differentiated to motor neurons (ATP7A-Cor1-MNs). In addition, as the reviewer suggested, we used a Isl1/2 marker to verify the motor neuron differentiation. Actually, we used MAP2 to determine neuritic maintenance and

microtubule networks in dendritic morphogenesis. In summary, we have added a scheme of differentiation protocol and immunostaining results of NPCs and motor neurons in **Figs. 3a-e** in the revised manuscript.

MN survival should be followed over a time course of degeneration.

Response: In our differentiation protocol, NPCs were generated from iPSCs through EB formation, and then NPCs were differentiated to MNs using SAG and RA (**Fig. 3a** and **method 10**). This protocol showed low efficiency of MN differentiation so that it was difficult to track MN survival. Another differentiation protocol using viral vector with fluorescence would make it easier to trace the MN survival, but we intended to avoid the viral vector protocol because we wanted the intact maintenance of patient-derived cells as possible. Instead, we performed various functional assays at specific time point (3 weeks after MN differentiation), including LDH level measurement, MTT assay and cleaved caspase-3 quantification assay which represents the level of cellular damage and viability (**Fig. 3h** and **Fig. S8**).

Another problem is the relatively large number of non-neuronal cells in the population, likely due to the use of FBS in the differentiation protocol. This can be a major confounding factor for downstream analysis related to survival and apoptosis. If the ALS line has a larger number of non-neuronal cells compared to the controls, it's hard to conclude that the LDH and other whole well assays specifically relate data on MN health.

Response: Thank you for the valuable comment. As mentioned above, however, NPCs differentiated from iPSCs were initially sorted by PSA-NCAM to eliminate non-neuronal population from rosettes as possible, minimizing the function of non-neuronal cells. We continuously checked neuronal cells state by Nestin staining (**Fig. 3b**).

The authors need to mention at what day in the differentiation process the cells were analysed as this can affect the results. Ideally, you want to conduct your assays when your MN are functional.

Response: To address the reviewer's concern, we have added a scheme of differentiation protocol (**Fig. 3a**) and mentioned time point of functional assay at each method section. We selected the time point of week 3, after confirming MN differentiation using a motor neuron marker (Isl1/2).

2. The n= values are a bit puzzling. When the authors mention n=20, are they really saying they did 20 independent differentiations? Or is the data an average of 20 wells? Are these technical replicates or biological? If these are independent differentiations, why do the numbers vary so widely? All 3 lines must be differentiated in parallel and compared at the

same time.

Response: We apologize for the insufficient explanation. It means technical replicates. From one corrected iPSC line (ATP7A-Cor1), we repeatedly differentiated it to neural progenitor cells (ATP7A-Cor1-NPCs) and further motor neurons (ATP7A-Cor1-MNs). All cells were differentiated in parallel and compared at the same time.

3. Since ALS affects MN, why have the authors chosen to analyse iNPCs instead of iNs? Without further characterization of the iNPC population, it is not advisable to use these for functional assays. Do the authors know whether the generated iNPCs are functional i.e. can they differentiate into neurons? Is the efficiency similar between control and ALS fibroblasts?

Response: The reviewer may raise a concern about using direct conversion technique to generate neurons (iNs) from the patient's fibroblasts. In this study, we intended to correct the ATP7A M1311V mutation via CRISPR-Cas9 and make a corrected line, so target cells should be proliferated sufficiently. Hence, we initially made corrected-iPC line and ultimately differentiated to motor neurons. In addition, obtained the patient-derived iNSCs (ATP7A-M1311V-iNSC) via direct conversion (**Fig. S10**) and compared them with control iNSCs from a healthy donor because of one potential concern that the pluripotency and differentiation of iPSCs might have affected the function of the resulting MNs. As a result, we confirmed that differentiated iNSCs were positive to a Tuj1 staining (**Fig. S10g**), indicating neurons' characteristics and no significant difference in reprogramming efficiency between control iNSCs and ATP7A-M1311V-iNSCs was observed by microscopic observation.

What is the significance of increasing LOX activity in response to Cu? Why was this done only for ALS iNPC?

Response: ATP7A is known for having a role in Cu transportation and lysyl oxidase (LOX) is a Cu-dependent enzyme. Therefore, we hypothesized that LOX activity is mainly dependent on the function of ATP7A and LOX activities for control-NPCs, ATP7A-M1311V-NPCs, ATP7A-Cor-NPCs. The LOX activity of ATP7A-M1311V-NPC was lower than control-NPC but rescued after gene correction, supporting our hypothesis (**Fig. S7a**). We further compared the LOX activities between Control-iNSC and ATP7A-M1311-iNSC, which showed similar tendency (**Fig. S11c** and **S11d**). We have added Cu-dependent experiments for both Control and ALS cell lines in **Fig. S11c** and **S11d** in the revised supplementary information as below.

4. The genetic data does not confirm that the ATP7A mutations causes ALS. Unless this can be confirmed in an independent cohort, we can only say this is likely associated with ALS.

Response: We agree with the reviewer’s comment. We have changed the word from ‘causation’ to ‘correlation in **line 140** and generally toned down the claim that ATP7A-M131V mutation has a potential responsibility for ALS overall in the revised manuscript.

5. Since the ATP7A mutation is not rare in Ashkenazi Jews and the authors claim this mutations as a cause of ALS, wouldn't we expect to find a higher incidence of ALS cases in this population. One possibility is that both alleles need to be mutated to cause ALS suggesting a recessive mode of inheritance. This will be in line with what the authors suggest that the mutation causes a loss of function. The authors need to support this line of inquiry by performing ATP7A1 knockdown and show impairment of MN function and/or survival.

Response: Thank you for the valuable comment. In this study, we showed that the ATP7A M1311V mutation is linked to ALS but this variant is not rare in the Ashkenazi Jewish population according to gnomAD. Because ALS is known to be a multifactorial disease and over 100 genes are intricately involved, we concluded that the ATP7A M1311V mutation can act as one crucial component in an ALS-related network and the ALS disease may be caused by a combination of other disease factors together with ATP7A M1311V at least for this patient. Recent studies [**references 51-56**] reported that common genetic variants can be risk variants in neurodevelopmental disorder patients, whereas rare mutations are found in normal tissue, supporting our conclusion that non-rare genetic variants can also be a

critical factor for genetic disease in a ALS patient.

In addition, from our trio WGS data, we found 5 candidate variants and assessed the ATP7A mutation at higher priority. The additional Sanger sequencing data revealed that ATP7A located on the X chromosome had a mutated variant only, identified to be hemizygous because the patient is male (**Fig. 1, Fig. S2 and lines 107-114**). As the review mentioned, we also think that the ATP7A mutation has a recessive mode of inheritance. Investigation of other genes' mutations or independent cohort study will be necessary for confirm it but, we believe that it would be a following study.

6. Table S4 not explained well.

Response: In this study, we used 2 different algorithms for network-based prioritization, which is described in **method 4**. One is a direct method (naïve Bayes; now rearranged to **Table S2**) and the other is an indirect method (Gaussian smoothing, now rearranged to **Table S2 Table S3**). We have added more description in **Line 98** in the revised manuscript.

7. Fig. 2c not explained in the text.

Response: We apologize for the insufficient explanation. This graph indicates the HDR efficiency increasing during isogenic cell line generation. We have added more detailed description about it in **lines 146-149** and detailed protocol for clump passage in **method 8**.

8. Fig S4C. I am guessing mutation in the PAM is synonymous but this should be mentioned.

Response: We have added the sentence 'The PAM sequence was disrupted by a synonymous mutation.' in **Fig. S5c** in the revised manuscript.

9. Please clarify what you mean by "5 repeated clump passages".

Response: Usually, a term 'clump passages' indicates iPS subculture passages because iPS grows as a colony formation. Hence, '5 repeated clump passages' means that 5 subcultures were undergone to generate isogenic cell line (**Fig. 2c**). Detailed protocol is noted in **Method 8**; 'Establishment of gene-corrected isogenic induced pluripotent stem (iPS) cell lines'.

10. Why was Cor2 dropped from further analysis? Is due to the fact that it showed higher ROS levels?

Response: Yes. When we measured the LOX activity and ROS level for 2 corrected lines,

ATP7A-Cor2-NPC lines showed higher variations in ROS level in ATP7A-Cor2-NPC, which might be due to cell-to-cell variation, thus we chose ATP7A-Cor1 line and repeatedly differentiated it to NPCs (ATP7A-Cor1-NPCs) and further MNs (ATP7A-Cor1-MNs).

REVIEWERS' COMMENTS:

Reviewer #1 (Remarks to the Author):

The authors addressed my concerns and made necessary changes in their manuscript. In this format, I do recommend the study to be published.

Reviewer #2 (Remarks to the Author):

The authors have addressed criticisms adequately.

Reviewer #3 (Remarks to the Author):

I do not agree that a decrease in MAP2+ cells at a single time point can be used as evidence of degeneration. But I agree with the authors that the other assays they have performed indicate a defect in survival. I think the manuscript should be amended to include this change. The authors can use the MAP2 data to indicate differences in neurite complexity.

Ideally, statistical analysis should be performed on biological replicates to get a true picture of the total variation. But it is important to mention how many independent differentiations were performed per experiment. For example, the authors might mention that they used n=24 technical replicates across 3 independent differentiations.

Otherwise, I feel the authors have addressed my other comments.

Reviewer #3 (Remarks to the Author):

I do not agree that a decrease in MAP2+ cells at a single time point can be used as evidence of degeneration. But I agree with the authors that the other assays they have performed indicate a defect in survival. I think the manuscript should be amended to include this change. The authors can use the MAP2 data to indicate differences in neurite complexity.

Response: We have revised the description of MAP2 staining assay result with that it is an evidence of neurite complexity as the reviewer commented (**line 166 or 171 with all contents shown**). We have also added a sentence in last paragraph to note the differences in cell viability between ALS patient-derived and ATP7A gene-corrected cells (**line 191-193 or 197-199 with all contents shown**).

Ideally, statistical analysis should be performed on biological replicates to get a true picture of the total variation. But it is important to mention how many independent differentiations were performed per experiment. For example, the authors might mention that they used n=24 technical replicates across 3 independent differentiations.

Response: In the previous rebuttal letter, we misunderstood the exact meaning of technical and biological replicates reversely. We would emphasize that all experiments were biologically replicated, that is, we repeated same assays for multiple samples of the same type of cells. We have further added the description about it in 'Statistical analysis and Reproducibility' of Methods section as editor requested, as below.

"In cases of using NPCs and MNs, all experiments were biologically replicated across 2 independent neural differentiations from iPSCs. That is, from one corrected iPSC line (ATP7A-Cor1), we repeatedly differentiated it to neural progenitor cells (ATP7A-Cor1-NPCs) and further motor neurons (ATP7A-Cor1-MNs). In case of using iNSCs, all experiments were reproduced at least 4 times. All attempts for replication were successful."